# Architecture of the ring formed by the tubulin homologue FtsZ in bacterial cell division

**Piotr Szwedziak†, Qing Wang†, Tanmay A M Bharat, Matthew Tsim, Jan Löwe\***

Structural Studies Division, MRC Laboratory of Molecular Biology, Cambridge, United Kingdom

**Abstract** Membrane constriction is a prerequisite for cell division. The most common membrane constriction system in prokaryotes is based on the tubulin homologue FtsZ, whose filaments in *E. coli* are anchored to the membrane by FtsA and enable the formation of the Z-ring and divisome. The precise architecture of the FtsZ ring has remained enigmatic. In this study, we report three-dimensional arrangements of FtsZ and FtsA filaments in *C. crescentus* and *E. coli* cells and inside constricting liposomes by means of electron cryomicroscopy and cryotomography. In vivo and in vitro, the Z-ring is composed of a small, single-layered band of filaments parallel to the membrane, creating a continuous ring through lateral filament contacts. Visualisation of the in vitro reconstituted constrictions as well as a complete tracing of the helical paths of the filaments with a molecular model favour a mechanism of FtsZ-based membrane constriction that is likely to be accompanied by filament sliding.

## Introduction

Membrane dynamics during cytokinesis are some of the most fundamental processes in biology, yet are poorly understood at the molecular and mechanistic level. During prokaryotic cell division the cell membrane and the cell envelope constrict, eventually leading to cell separation. In most bacteria and archaea, this is guided by a ring structure containing the bacterial tubulin homologue FtsZ protein (*Bi and Lutkenhaus, 1991*; *Löwe and Amos, 1998*), which polymerises in a GTP-dependent manner (*Mukherjee and Lutkenhaus, 1994*). During constriction, the FtsZ ring decreases in diameter through an unknown mechanism. The C-terminal tail of FtsZ links it to other components of the divisome, an ensemble of many proteins that facilitates essential functions during the cell division process, most importantly remodelling of the cell envelope. Components of the divisome engage in cell wall synthesis (PBPs), synchronisation with chromosome dimer resolution (FtsK), lipid II cell wall precursor flipping (FtsW or MurJ), and many components currently have no known function (reviews: *Adams and Errington, 2009*; *Lutkenhaus et al., 2012*).

In *Escherichia coli*, binding of the FtsZ tail to ZipA and possibly more importantly to FtsA anchor the FtsZ ring to the membrane (*Pichoff and Lutkenhaus, 2007*). FtsA is a bacterial actin-like protein that forms domain-swapped, canonical actin-like protofilaments that are membrane associated through FtsA's C-terminal amphipathic helix (*Pichoff and Lutkenhaus, 2005*; *Szwedziak et al., 2012*; *van den Ent and Löwe, 2000*). Several cellular regulatory processes influence the onset and progression of cell division through mechanisms that directly act on FtsZ. For example, SulA is induced during the SOS stress response and sequesters monomers, stopping FtsZ polymerisation (*Chen et al., 2012*). In *E. coli*, both nucleoid occlusion and the oscillating, pole-protecting MinCDE system contain components that inhibit FtsZ function within the ring directly (*Bernhardt and de Boer, 2005*; *Dajkovic et al., 2008*).

Although progress has been exhilarating over that past 20 years or so, some of the most fundamental questions still remain: what happens during FtsZ ring constriction? How are the filaments arranged in the ring? What drives constriction?

**\*For correspondence:** jyl@mrc-lmb.cam.ac.uk

†These authors contributed equally to this work

**Competing interests:** The authors declare that no competing interests exist.

**Reviewing editor**: Werner Kühlbrandt, Max Planck Institute of Biophysics, Germany

**eLife digest** Cell division is the process by which new cells are made. It is therefore vital for the growth and development, and the regeneration and repair of damaged tissues. When bacterial and animal cells divide, they must constrict their membrane inwards to split a single cell into two. In most bacteria, this constriction is guided by a ring-like structure that contains filaments of a protein called FtsZ. During cell division, this structure forms around the inside edge of the cell and when it contracts, it pulls the membrane inwards and causes the cell to constrict and eventually divide.

In recent years, this arrangement of FtsZ filaments has been intensively investigated, giving rise to various theories about how it is made and how it works: for example, some recent studies suggested that FtsZ does not form a continuous ring. Nevertheless, many details about the cell division process remain unknown.

Szwedziak, Wang et al. have now investigated this protein ring in two species of bacteria by turning to advanced forms of microscopy to closely observe its structure and how it works. This included mapping the ring in three dimensions. Contrary to earlier reports that the FtsZ ring is discontinuous, in both a bacterium called *Caulobacter crescentus* and another called *Escherichia coli,* the ring forms a continuous shape made up of overlapping filaments.

Szwedziak, Wang et al. then increased the levels of two of the ring's main components: the FtsZ protein that forms the filaments and a protein that anchors these filaments to the cell membrane. This caused the modified cells to constrict and divide at extra sites, which resulted in the formation of abnormally small cells. These findings suggest that these two ring components by themselves are able to generate both the structures and force required for cell constriction. This is supported by the fact that when they were introduced into artificial cell-like structures, these proteins spontaneously self-organised into rings and triggered constriction where they formed.

Szwedziak, Wang et al. propose that constriction only starts once the FtsZ protein forms a closed ring and that the ring's overlapping filaments slide along each other to further decrease its diameter and constrict the cell. The degree of filament overlap likely also increases with constriction, requiring filaments to be shortened to maintain sliding. This shortening, along with sliding, could provide a mechanism by which to drive the constriction process.

This work will be followed by even more detailed studies in order to understand the process of bacterial cell division at the atomic scale and how the cell's wall is reshaped during the process. In the long run, intricate knowledge of how a bacterial cell divides might enable the design of new classes of antibiotics targeting the molecular machinery involved.

Many different models have been proposed for the mechanism of FtsZ-based constriction (reviewed in *Erickson, 2009*; *Erickson et al., 2010*). Essentially, three different approaches have been taken to validate the models: in vivo imaging of FtsZ constrictions using fluorescently labelled proteins. Electron cryotomography of frozen hydrated cells without labelling and, thirdly, in vitro reconstitution experiments with pure, fluorescently labelled proteins. The most recent results emanating from those studies are that the rings appear to show strong fluorescence intensity variations that may suggest that the FtsZ ring is discontinuous (*Holden et al., 2014*). Equally, tomography data have been interpreted to show scattered individual FtsZ filaments, some precise distance away from the membrane (*Li et al., 2007*). Reconstitution experiments with FtsZ and FtsA showed dynamic behaviour and liposome constrictions (*Osawa and Erickson, 2013*; *Loose and Mitchison, 2014*). However, obtaining detailed molecular and mechanistic information regarding the FtsZ ring, particularly the arrangement of individual filaments and subunits within a constricting Z-ring, has remained a formidable challenge.

In this study, we obtained high-resolution images of the FtsZ ring in *Caulobacter crescentus* and *Escherichia coli* by means of electron cryotomography. Furthermore, we reconstituted a minimal constriction force-generating system from purified components in vitro, encapsulating *Thermotoga maritima* FtsA (TmFtsA) and FtsZ (TmFtsZ) in liposomes of sizes corresponding to those of a bacterial cell. We produced images and three-dimensional maps of filaments arranging themselves into ring structures around the liposome perimeters that coincided with constriction sites. The observed FtsZ ring architectures in *C. crescentus* and *E. coli* cells and in liposomes favour a mechanism of FtsZ-based cell membrane constriction that is accompanied by filament sliding, as was proposed previously (*Lan et al., 2009*).

## Results

### Single-layered and continuous FtsZ ring in unmodified *C. crescentus* cells

We started out by visualising division sites in an unmodified *C. crescentus* strain (NA1000/CB15N) because the thin *Caulobacter* cells are most suitable for electron cryotomography. When a log-phase culture was plunge-frozen and imaged, many dividing cells could be found. At the division sites, a series of dots arranged in a single line were found (*Figure 1A*, top). Careful analysis of cellular tomograms (*Video 1* and *Figure 1A*, bottom) revealed that the dots were in fact 2D projections of filamentous structures encircling the cell and likely forming a continuous ring, disrupted in the images at the top and bottom by the missing wedge of the tomography method. The filaments were at a distance of 15 ± 2 nm from the inner membrane (*Figure 1B*), as previously reported (*Li et al., 2007*). The long-standing problem of the missing wedge in electron tomography caused by our current inability to tilt the specimen much beyond 65° (*Figure 1A*, bottom, white triangle, see also *Figure 1—figure supplement 1* for more details on the missing wedge problem related to this study) makes it impossible to follow features all the way around the cell's perimeter. It is important to note, however, that the protein filaments are visible and uninterrupted everywhere the missing wedge allows it, as can be gauged from the disappearance of the cell membrane and envelope (*Figure 1A,B*). We could detect the filamentous rings in 20 out of 28 dividing cells after tomography. Of the 8 without obvious filamentous structures, the division had progressed too far in 3 and 5 were of poor quality because of the cell's orientation with respect to the tilt axis (*Figure 1—figure supplement 1*). Hence, the finding of complete rings is supported by the fact that perfect coincidence of any hypothetical gaps in constricting rings with the missing wedges in all of the analysed tomograms, found after all in entirely random orientations, would be remarkably implausible.

### Identical FtsZ ring architecture in unmodified *E. coli*

To investigate the generality of these findings, we imaged unmodified *E. coli* B/r H266 cells, which we chose because of their thinness (*Woldringh, 1976*). Although the *E. coli* cells were thicker than *C. crescentus*, we found the wild-type filaments still discernible, most obviously when imaged along the long axis of cells (*Figure 1C* and *Figure 1—figure supplement 2*, especially B and D; *Video 2*), and these were 16 nm away from the IM. Based on these observations we concluded that *E. coli* Z-rings were, like in *C. crescentus*, probably continuous and consisted of single-layered bands that are 5–10 filaments wide.

In order to investigate if the filaments imaged in wild-type cells so far contained FtsZ protein, we over-expressed FtsZ(D212A), a mutant protein that hydrolyses GTP much more slowly (*Redick et al., 2005*) in *E. coli* B/r H266 cells. When over-expressed to 2.5-fold total FtsZ (*Figure 1D*), the protein formed a wide single layer of filaments at the division site (*Figure 1E*, see also *Video 3* and *Figure 1—figure supplement 3*), very similar to the bands seen in unmodified cells, but wider and containing more filaments as would be expected because there is now more FtsZ protein in the cell and filament dynamics have been reduced because of the GTPase-reducing mutation D212A. *Figure 1F* (*Figure 1—figure supplement 3A–C*) provides a view rotated by 90°, showing again that the filamentous ring was located approximately 16 nm away from the inner membrane (IM). The band of filaments most likely consisted of doublets of individual protofilaments, as is indicated in *Figure 1G*, which shows a cell with higher expression level (*Figure 1—figure supplement 3A–C*). The filaments were on average 6.8 nm apart (n = 17, distance between centres of adjacent filaments within a doublet, *Figure 1G*, lower). It is currently not known what lateral interactions between FtsZ filaments cause this arrangement or if it is facilitated by other proteins.

### The filaments observed in the tomograms are FtsZ

Because no specific label for electron cryotomography currently exists that works in *E. coli*, we decided to further confirm the identity of the filaments as being composed of FtsZ by systematic perturbations of the system in four (a–d) separate experiments with subsequent imaging by electron cryotomography (*Figure 1—figure supplement 5*, *Supplementary file 1A,B*). (a) Introducing extra amino acids into the flexible linker (*Buske and Levin, 2013*; *Gardner et al., 2013*) within FtsZ that separates the globular N-terminal domain of FtsZ from the small C-terminal helix that binds FtsZ's membrane anchor, FtsA (*Ma and Margolin, 1999*; *Szwedziak et al., 2012*), increased the distance

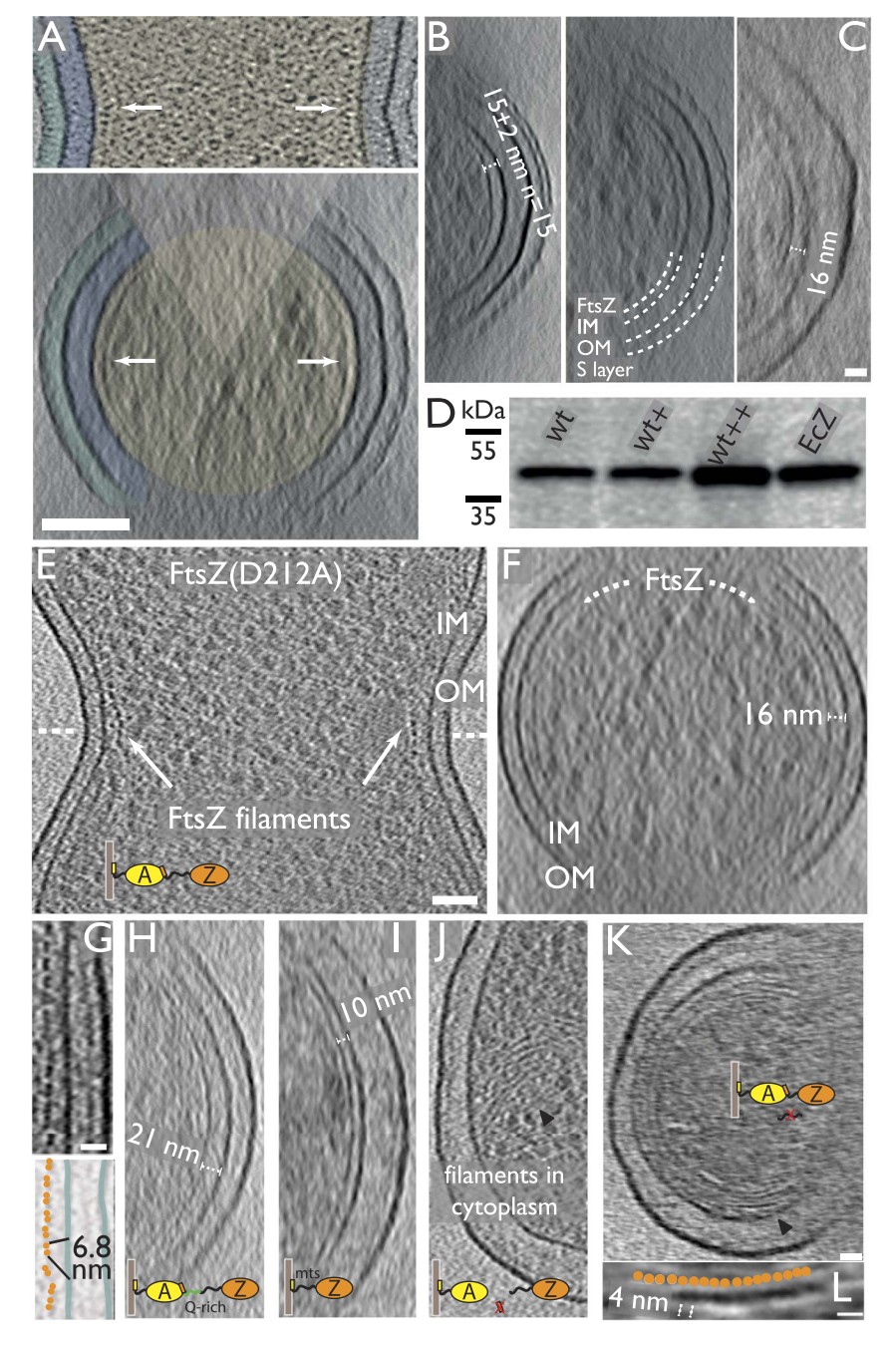

**Figure 1**. FtsZ forms bands of filaments completely encircling *C. crescentus* and *E. coli* division sites, as visualised by electron cryotomography. (**A**) *C. crescentus* NA1000/CB15N division site with filaments near the inner membrane IM (top panel, black dots highlighted by arrow, see also *Video 1*). Bottom panel shows the same cell rotated 90° around the short axis of the cell. The Z ring (arrow) is continuous and only invisible where there is no image because of the missing wedge (shaded triangle) (see *Figure 1—figure supplement 1* for more details on the missing wedge problem). The cytoplasm (beige), periplasm (blue), and space between the OM and S layer (cyan) have been coloured for clarity. (**B**) More examples of continuous FtsZ rings found in *C. crescentus* cells. The filaments were on average 15 nm from the inner membrane. (**C**) Electron cryotomographic slice of the constriction site of a B/r H266 *E. coli* cell visualised perpendicular to the longitudinal axis, showing very similar FtsZ filaments when compared to *C. crescentus* (*Figure 1A,B*) and FtsZ(D212A) expressing *E. coli* cells (*Figure 1F*) and having roughly the same distance (16 nm) to the IM. *Video 2* demonstrates the likely helical nature of the arrangement of *Figure 1. Continued on next page*

*Figure 1. Continued*

the FtsZ filaments (see also *Figure 1—figure supplement 2*). (**D**) Western blot showing total FtsZ levels in cells used in (**E**–**G**) are about 2.5× that of wild-type cells. (+) refers to un-induced, (++) was induced by 0.02% arabinose. EcZ is purified *E. coli* FtsZ protein. (**E**–**G**) 10-nm thick electron cryotomographic slices of *E. coli* cells expressing FtsZ(D212A) protein in a wild-type B/r H266 background. See also *Figure 1—figure supplement 3*. (**E**) *E. coli* division site showing the cross-section of FtsZ filaments (single row of black dots) at the constriction site. See *Video 3*. (**F**) Visualisation of the same cell along the longitudinal axis shows that FtsZ filaments are located ~16 nm from the inner membrane (IM). (**G**) Closer examination of the constriction site of another cell with higher expression level reveals FtsZ filaments form pairs, appearing as doublets of dark dots (upper) and orange spheres in the schematic illustration, on average 6.8 nm apart within the doublets (lower). (**H**–**K**) 10-nm thick electron cryotomographic slices of *E. coli* cells expressing engineered protein constructs based on FtsZ(D212A) (see also *Figure 1—figure supplements 3,5* and *Supplementary file 1, Table B*). (**H**) Extending the C-terminal linker of FtsZ by inserting a linker sequence pushes the filaments further away from the IM (distance changed from 16 nm to a somewhat variable 16–21 nm). (**I**) Replacing the C-terminal FtsA-binding sequence of FtsZ with a membrane-targeting sequence (mts) makes FtsZ directly bind to the IM and results in FtsZ filaments closer to IM (distance changed from 16 nm to 10 nm). No cell constrictions were observed with this construct. (**J**) Removing the C-terminal FtsA-binding sequence of FtsZ renders it unable to maintain a fixed distance to the IM and FtsZ filaments that were observed within the cytoplasm. (**K**) Removing the C-terminal flexible linker of FtsZ makes it prone to form multiple layers of filaments that form complete rings or helices. Tomography using this construct works better because it produces small minicells. (**L**) A closer inspection of the area marked with the black arrowhead in G shows beads along the filament as illustrated by the schematic drawing with a repeat distance of 4 nm as expected for FtsZ filaments. IM: inner membrane; OM: outer membrane; WT: wild-type; Q-rich: FtsN-derived flexible linker; mts: membrane-targeting sequence. Scale bars: 100 nm in (**A**) and (**B**), 50 nm in (**E**, **F**, **H**, **I**, **J**), 20 nm in (**C**, **G**, **K**), 10 nm in (**H**), 20 nm in (**L**).

The following figure supplements are available for figure 1:

**Figure supplement 1**. The missing wedge problem in cellular electron cryotomography.

**Figure supplement 2**. Electron cryotomograms of wild-type *E. coli* cells show filaments at the constriction sites.

**Figure supplement 3**. FtsZ forms bands of filaments at constriction sites in *E. coli* cells.

**Figure supplement 4**. Engineered FtsZ proteins form filaments with altered localisation patterns in *E. coli* cells.

**Figure supplement 5**. Overview of FtsZ constructs used for in vivo tomography.

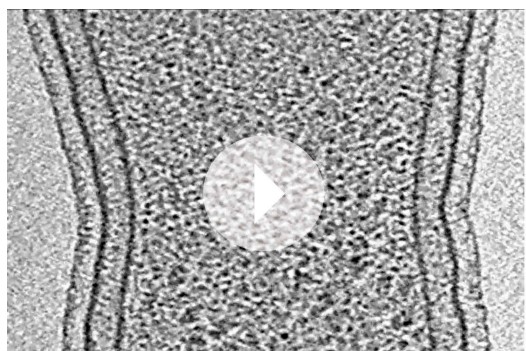

**Video 1**. Tomogram of a wild-type *C. crescentus* cell showing tomographic slices parallel to the longitudinal axis of the cell. A single layer of dark dots corresponding to cross-sections of FtsZ filaments is clearly visible at a distance from the membrane on both sides of the septum. The missing wedge is located at top and bottom. The distance between adjacent filaments highlighted by the arrow varies along the z-direction. This corresponds to *Figure 1A*.

between the FtsZ ring and the IM from 16 nm to a somewhat variable 16–21 nm (*Figure 1H* and *Figure 1—figure supplement 4A–C*). (b) Removing the C-terminal FtsA-interacting helix and replacing it with a membrane-targeting sequence (mts) from MinD protein (*Hu and Lutkenhaus, 2003*) shortened the distance from the IM to 10 nm. No constrictions of the cells were observed (*Figure 1I* and *Figure 1—figure supplement 4D*). Therefore, despite it having been used in earlier studies (*Osawa et al., 2008*, *2009*; *Osawa and Erickson, 2011*; *Loose and Mitchison, 2014*), we agree with previous findings (*Osawa et al., 2008*) that this construct is non-functional in vivo and we did not include it in our subsequent in vitro investigations below. (c) Removing the C-terminal FtsA-interacting helix from FtsZ detached the filaments from the membrane, making them appear throughout the cytoplasm (*Figure 1J* and *Figure 1—figure supplement 4E*). (d) Finally, removing most amino acids between

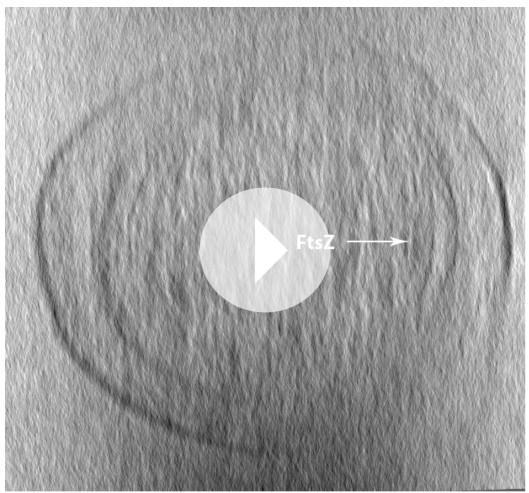

**Video 2**. Tomogram of a wild-type *E. coli* cell showing the constriction site along the longitudinal axis of the cell. FtsZ filaments are visible in certain slices and are likely to be forming continuous helices indicated by its pattern when viewed along the slices. This corresponds to *Figure 1C*.

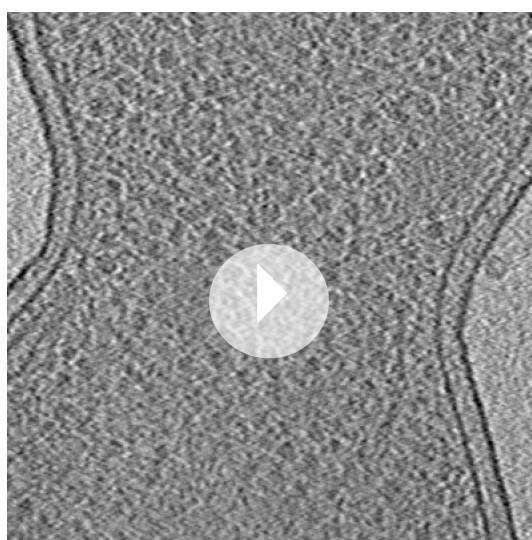

**Video 3**. Tomogram: FtsZ(D212A) expressed in *E. coli* cell forms doublet FtsZ filaments at the constriction site. The video shows tomographic slices parallel to the longitudinal axis of the cell. One single layer of dark dots corresponding to cross-sections of FtsZ filaments is clearly visible, and these dark dots tend to form pairs suggesting a doublet FtsZ filament architecture at the constriction sites formed with FtsZ and FtsZ(D212A). This corresponds to *Figure 1E*.

the C-terminal FtsA-interacting helix and the globular body of FtsZ caused a minicell phenotype. Because of their size, minicells produce tomograms of higher quality and it was possible to determine the longitudinal subunit repeat of the filaments to be around 4 nm, very close to the expected value of 4.2 nm for FtsZ (*Figure 1K and L* and *Figure 1—figure supplement 4G*) (*Erickson et al., 1996*). We conclude that the filament localisations reacted to our perturbations as expected for FtsZ and the subunit repeat was the same as for all known FtsZ protofilaments. The minicell tomogram (*Figure 1K*) is another indication that the filaments most likely encircle entire cells.

## Extra septa generated by additional FtsZ and FtsA function in cell separation

Encouraged by reports that simultaneous overexpression of FtsZ and its membrane anchor FtsA led to additional division sites (*Begg et al., 1998*), we imaged *E. coli* cells in which extra FtsZ(D212A) and FtsA were produced from a bicistronic expression vector, by electron cryotomography (*Figure 2A–E*). Providing just these two proteins in excess (twofold to fourfold total vs WT) produced a severe phenotype with many extra constrictions visible (*Figure 2A,B*). Since in these cells the normal FtsZ to FtsA ratio had been altered to be close to 1:1, from normally 5 FtsZ:1 FtsA (*Rueda et al., 2003*), FtsA filaments became visible at the constricting division sites (*Figure 2C–E*). Actin-like FtsA binds to the membrane directly via its C-terminal amphipathic helix and polymerises into canonical actin-like protofilaments (*Lara et al., 2005*; *Szwedziak et al., 2012*). In this way, an artificially strong 'FtsA ring' became apparent, indicating that FtsA is located between the IM and FtsZ, being 8 nm away from both (*Figure 2D,E*).

We then demonstrated that the FtsZ(D212A) and FtsA-over-expressing *E. coli* cells, we had imaged at high-resolution with electron cryotomography, were dividing and separating, despite looking quite distorted. For this, we employed structured illumination microscopy (SIM) on live cells (*Figure 2F*). The cells showed a strong minicell phenotype, while performing many cell divisions randomly distributed along the cell length. This indicated to us that the extra constrictions and division sites were functional in the sense that they led to complete cell separation (abscission).

We concluded that the ability to produce extra constriction sites and divisions by just providing more FtsZ and FtsA indicates that these two proteins may be central components of the IM constriction force generator that is localised within the inner division apparatus (*Rico et al., 2013*) and that it

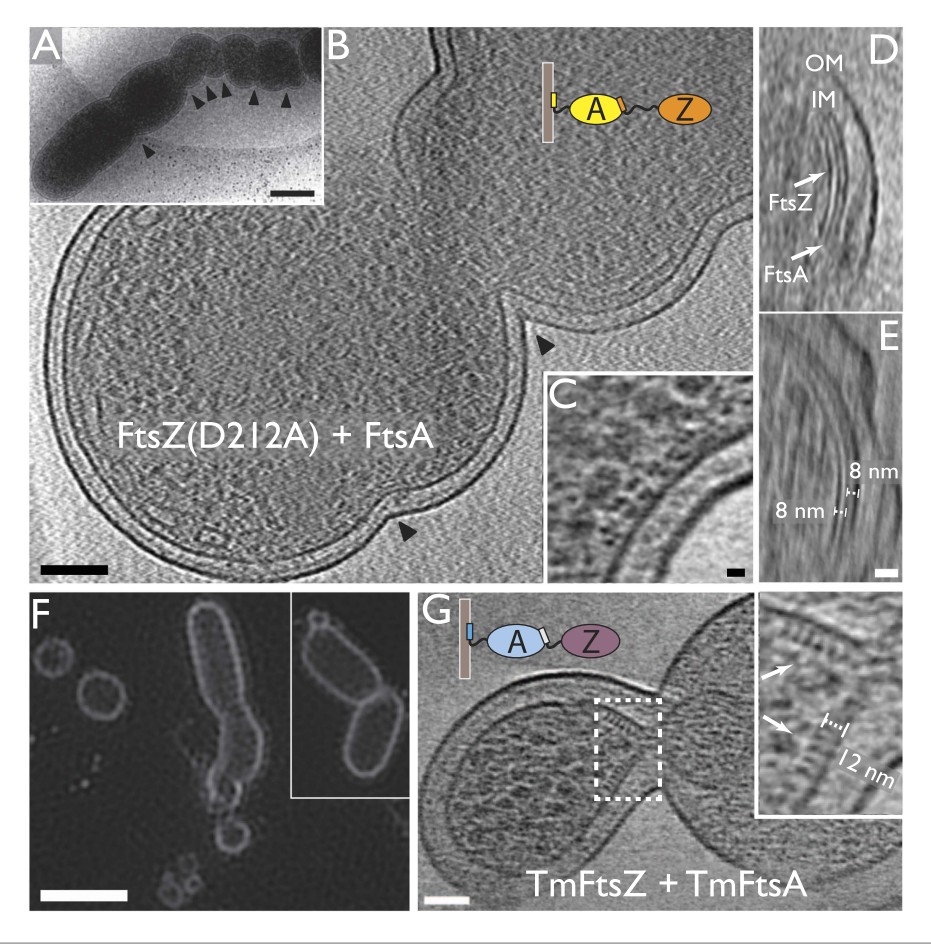

**Figure 2**. Co-expression of FtsZ and FtsA in *E. coli* cells leads to extra septa. (**A**) A low-magnification 2D electron cryomicrograph (transmission) showing multiple constriction sites (marked with black arrowheads) along the cell. (**B–E**) 10-nm thick electron cryotomographic slices of cells co-expressing FtsZ(D212A) and FtsA (bicistronic, 1:1). Two layers of dots are visible at constriction sites in (**B**) and (**C**), corresponding to FtsZ filaments and FtsA filaments, respectively, as labelled in the orthogonal view along the long axis of the cell (**D**). FtsA filaments are almost in the middle between FtsZ filaments and the IM, at a distance of 8 nm from both FtsZ filament and IM as indicated in (**E**). (**F**) Structured illumination microscopy images of cells expressing FtsZ(D212A) and FtsA, showing cell division and minicell formation, proving that the extra septa function to completion. (**G**) 10-nm thick electron cryotomographic slice of an *E. coli* minicell formed from cells expressing *Thermotoga maritima* FtsZ and FtsA proteins, with a deeply constricted area showing cross-sections of FtsZ and FtsA filaments (black dots marked with white arrows). Distance between FtsZ filaments and IM is around 12 nm (inset in **G**). The view highlights striking similarities to the in vitro reconstruction shown in *Figure 3H–J & 5C*. IM: inner membrane; OM: outer membrane. Scale bars: 500 nm in (**A**), 100 nm in (**B**), 10 nm in (**C**, and also for inset in **G**), 20 nm in (**E**, and also for **D**), 2 μm in (**F**).

may be possible to reconstitute membrane constriction with just the two of them. Indeed, this has recently been reported and has been imaged at low resolution using fluorescence microscopy (*Osawa et al., 2008*; *Osawa and Erickson, 2013*), although not providing any molecular insights.

## Reconstituting liposome constrictions in vitro using *T. maritima* FtsA and FtsZ

So, we then used purified FtsZ and FtsA proteins for in vitro reconstitution experiments, in order to observe constriction. We avoided fluorescently tagged proteins as they have been shown to introduce artefacts (*Margolin, 2012*). We therefore used completely unmodified TmFtsZ and TmFtsA proteins from *Thermotoga maritima*, both of which are easy to obtain and handle and have crystal structures available (*Oliva et al., 2004*; *van den Ent and Löwe, 2000*). It should be noted that extra care had to

be taken in order to obtain proteins that did not have their disordered but important C-terminal tails cleaved during purification. Just to confirm that TmFtsZ and TmFtsA formed structures similar to the *E. coli* counterparts in vivo, we over-expressed TmFtsZ and TmFtsA in *E. coli* and imaged the sample by electron cryotomography (*Figure 2G*). Minicells were formed and, at constriction sites, they contained filaments that closely resembled the filament arrangement observed here for *E. coli* FtsA and FtsZ over-expression in *E. coli* (*Figure 2C*). The distance of TmFtsZ to the IM was shorter at 12 nm; this was expected because the linker between the very C-terminal TmFtsA-interacting helix and the body of TmFtsZ is much shorter, measuring around nine amino acids. Minicell formation might indicate that TmFtsA and TmFtsZ interacted with the *E. coli* cell division machinery or even supported membrane constriction on their own, but we did not investigate this further.

## FtsA and FtsZ form spirals on a flat lipid surface

When added onto a flat lipid monolayer, TmFtsZ and TmFtsA formed striking spirals (*Figure 3A*). The filaments forming the spirals tended to form weak doublets and in the centre of the spirals, white material was visible that may have been lipid that had been pushed up by the spiral constricting, possibly via a sliding filament mechanism as has previously been observed for FtsZ alone by AFM (*Mingorance et al., 2005*). Intriguingly, much larger dynamic chiral spirals of polar FtsA and FtsZ filaments have recently been reported on supported lipid bilayers (*Loose and Mitchison, 2014*), but the exact relationship with our observation is currently unclear as treadmilling and no constriction were observed on the supported bilayers.

## FtsA and FtsZ polymers together generate negative curvature on liposome surfaces

Since the FtsZ ring in vivo does not act on flat membranes, we then switched to liposomes formed from *E. coli* lipid extract. First, TmFtsZ and TmFtsA were added to pre-formed liposomes so that the proteins remained on the outside (*Figure 3B* and *Figure 3—figure supplement 1A*). Reactions containing liposomes and proteins, as indicated, were vitrified and imaged by conventional 2D transmission electron cryomicroscopy. When no protein was added, the liposomes appeared as almost perfect circles (spheres in projection) and the bilayers were clearly visible as a double line, 5 nm apart (*Figure 3B*, top, *Figure 3—figure supplement 2D*). The addition of TmFtsA alone led to the formation of an additional layer, probably consisting of only partly polymerised protein, and no strong deformations were observed. TmFtsZ alone did not cause deformations or generation of an additional layer (*Figure 3—figure supplement 2B*). But when both TmFtsA and TmFtsZ were added, two extra layers, in addition to the liposome bilayer, became visible (*Figure 3B*). Particularly in the presence of nucleotide, strong negative curvature was induced, leading to deformations of the liposomes (*Figure 3—figure supplement 1A*). As was suggested previously, the co-polymerisation of FtsZ and FtsA will lead to bending and curvature because the subunit repeat lengths of FtsZ and FtsA are roughly 4 and 5 nm, respectively (*Figure 3—figure supplement 1B* and *Figure 3—figure supplement 2A*) (*Szwedziak et al., 2012*). We propose that the subunit repeat mismatch causes some deformations from the outside of the liposomes, especially when FtsZ polymerisation-inducing GMPCPP is present so that long filaments are formed that will exert more mechanical force.

## Incorporating FtsA and FtsZ on the inside of liposomes leads to spontaneous constrictions

Since TmFtsZ and TmFtsA induce negative membrane curvature, we concluded that in order to reconstitute the actions of these proteins correctly, we needed to incorporate them on the inside of liposomes. For this, CHAPS detergent-solubilised *E. coli* lipid extract was mixed with the proteins at high concentrations and then diluted many-fold. Lowering of the detergent concentration by dilution led to spontaneous liposome formation with most of the proteins on the inside. We demonstrated with a time-lapse experiment that liposomes did not form around pre-existing FtsZ scaffolds since we observed that most liposomes were initially perfectly spherical and that they deformed over a 30-min period, after which most of them were heavily misshapen (*Figure 3—figure supplement 2C,E*).

The liposomes were then analysed by transmission electron cryomicroscopy. Amazingly, when both TmFtsZ and TmFtsA were included on the inside of liposomes, clear constriction sites appeared and these occurred only when supported by filaments (*Figure 3C–G*). The liposomes were around 300 nm in diameter, similar to that of a small bacterial cell. The protein filaments were arranged into three

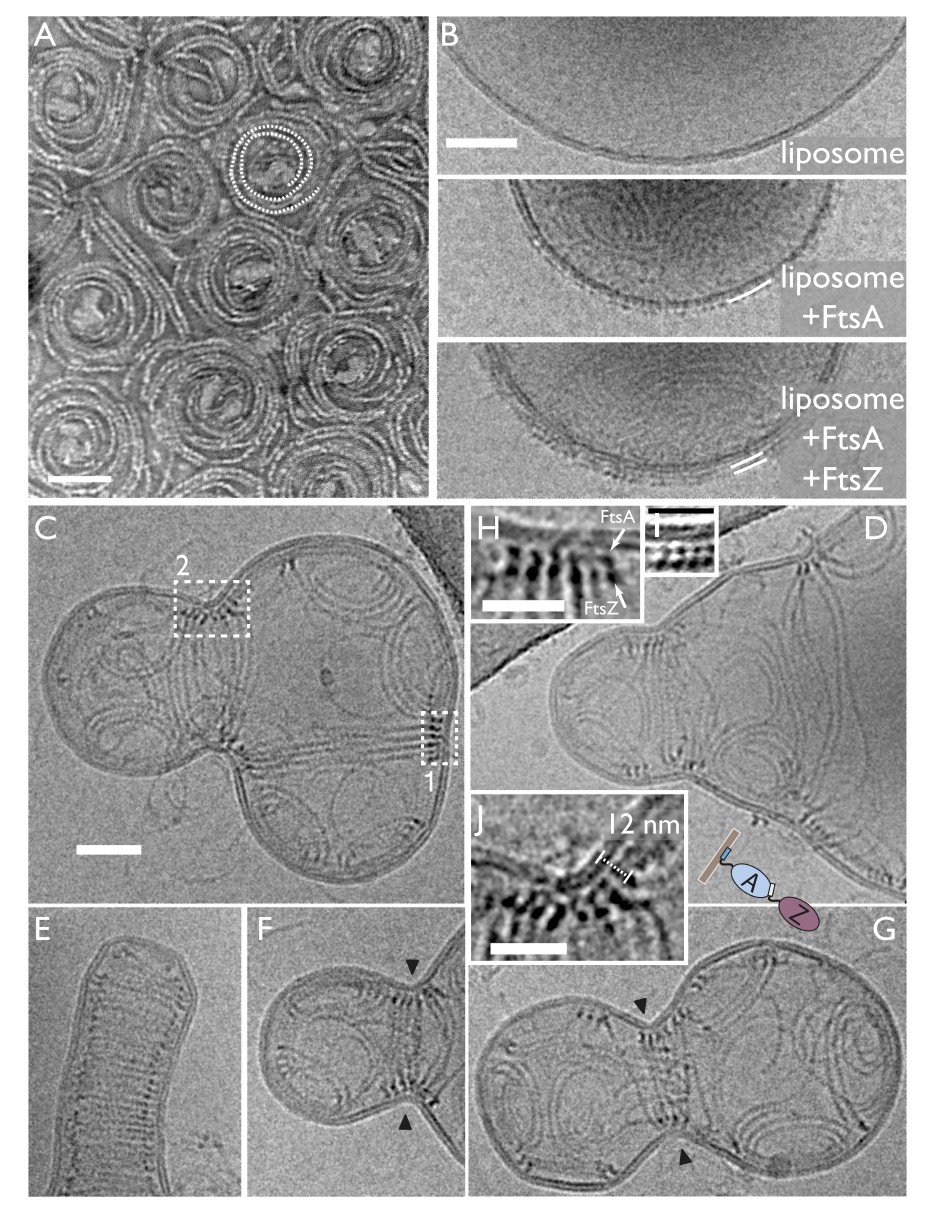

**Figure 3**. In vitro reconstitution of bacterial cell membrane constriction by the FtsZ ring from purified components. (**A**) *Thermotoga maritima* FtsA (TmFtsA) and *Thermotoga maritima* FtsZ (TmFtsZ) form spirals on a flat lipid monolayer, as indicated by a white dotted line. The filaments tend to appear as double strands (doublets). Negative-stain electron microscopy. (**B**) Transmission electron cryomicroscopy allows resolution of the inner and outer leaflet of undisturbed liposomes (top panel). When TmFtsA is added to the outside, an additional layer of density corresponding to FtsA becomes apparent (middle panel). Recruitment of TmFtsZ by TmFtsA leads to the formation of two layers (bottom panel). Taken together, we conclude that FtsA is sandwiched between the membrane and FtsZ filaments (bottom panel). See also *Figure 3—figure supplement 1* and *Figure 3—figure supplement 2*. (**C–G**) Constriction sites are efficiently formed when TmFtsA and TmFtsZ are encapsulated in liposomes that have sizes comparable to bacterial cells. Five representative liposomes are shown using transmission electron cryomicroscopy (hence are 2D projections of 3D objects). Importantly, constriction sites are only formed where a ring made of the two proteins is present (black arrowheads) and not at other sites where filaments are located. The TmFtsA and TmFtsZ layers are clearly visible (inset **H**, same as boxed area '1' in **C**; inset **J**, same as boxed area '2' in **C** and inset **I**, which is from *Figure 4* electron cryotomography data) and the protein's organisation mirrors that present in *E. coli* cells (compare with *Figure 2C*). The distance of 12 nm between TmFtsZ and the

*Figure 3. Continued on next page*

*Figure 3. Continued*

membrane (inset **J**) resembles that found in over-expressing cells (see *Figure 2G* and also *Figure 5C*). (**E**) Intriguingly, liposomes are being constricted (partially) in the absence of added nucleotide. Scale bars: 50 nm in (**A–C**), 25 nm for insets.

The following figure supplements are available for figure 3:

**Figure supplement 1**. TmFtsZ and TmFtsA on the outside of liposomes and in the presence of GMPCPP deform liposomes.

**Figure supplement 2**. Control experiments showing that both TmFtsA and TmFtsZ form straight filaments when polymerised separately. And liposomes deform mostly after dilution.

distinct structures: arcs that were mostly single filaments, presumably formed by the co-polymerisation of TmFtsZ and TmFtsA, showing the characteristic curvature caused by the repeat mismatch (*Szwedziak et al., 2012*); spirals with decreasing curvature that appeared in perfectly spherical areas of the liposomes; and rings of filaments, appearing as bands in projection, formed around liposome constriction sites, with varying diameters. It is very important to note that constrictions only appeared where there were rings of filaments. Filaments within the liposomes had very different curvatures, for example in *Figure 3D*, filaments seemed to go round the liposome at a very large diameter, compared to the ones in the constriction zone further to the left.

When the constriction sites were imaged at higher magnification, it became possible to discern the TmFtsA and TmFtsZ filaments end-on (*Figure 3H–J*). The FtsA filaments were again sandwiched between the liposome membrane and the FtsZ filaments, just as in the images obtained with *E. coli* cells (*Figure 2C,G*). The architecture is easily explained with TmFtsA-binding to the inside of the liposome membrane via its amphipathic helix, presumably polymerising, and FtsZ polymerising on top of FtsA, binding to it via its C-terminal FtsA-binding peptide (*Pichoff and Lutkenhaus, 2005*; *Szwedziak et al., 2012*). Nucleotide presence had some influence on the appearance of these constricted liposomes as GTP addition produced the most bilobed liposomes. Constriction itself, however, was largely nucleotide hydrolysis independent since not adding any nucleotide produced constrictions as well, and they appeared even tighter (*Figure 3E*). Therefore, despite not being strictly required for constriction, we believe that the nucleotide only has an influence on the appearance thereof and this might be due to the fact that FtsZ forms much longer filaments with polymerisation-inducing GTP added and this would enable the filaments to span larger liposomes when the process starts. Given that TmFtsZ is a hyperthermophilic protein, significant hydrolysis of GTP to GDP is not expected.

## Detailed architecture of the liposome constrictions revealed in three dimensions by electron cryotomography

Next, we employed electron cryotomography to image the liposomes in three dimensions (*Figure 4*, *Figure 4—figure supplement 1*, *Videos 4–10* ). Because the samples only contained lipid and two proteins, contrast was very high in the resulting tomograms (*Video 4 and 5*), making it possible to represent the volume data without segmentation, as single-threshold surfaces or as volume renderings (*Video 6–10*, *Figure 4—source data 1* PyMOL session file). None of the figures or videos we present has been segmented, manually or automatically. *Figure 4A*, top shows a constricted liposome in stereo, highlighting the three distinct filament architectures in detail: arcs, spiral domes at the 'poles', and the filamentous ring, pulling and constricting the membrane. Note that the liposome was only deformed where the ring was located (and where it unfortunately touched the carbon grid at the top right). More examples in *Figure 4A*, bottom and *Figure 4—figure supplement 1* show the same overall architecture, with the same mix of filament architectures (also *Videos 6–10*). The TmFtsAZ rings were between 30 and 90 nm in diameter, when we looked at several different liposomes, and the filaments were on average 7.8 nm apart (n = 16) laterally (*Figure 4B,C*) as compared to 6.8 nm seen in *E. coli* cells (within doublets). Because contrast was very high, the ring of filaments could be traced in most tomograms all around the inside of the liposome and this revealed that the filaments were not totally equidistant and often came into contact. This was also true for the in vivo situation in *C. crescentus* (*Video 1*) where the distance between the filaments (grey arrow) changes around the ring.

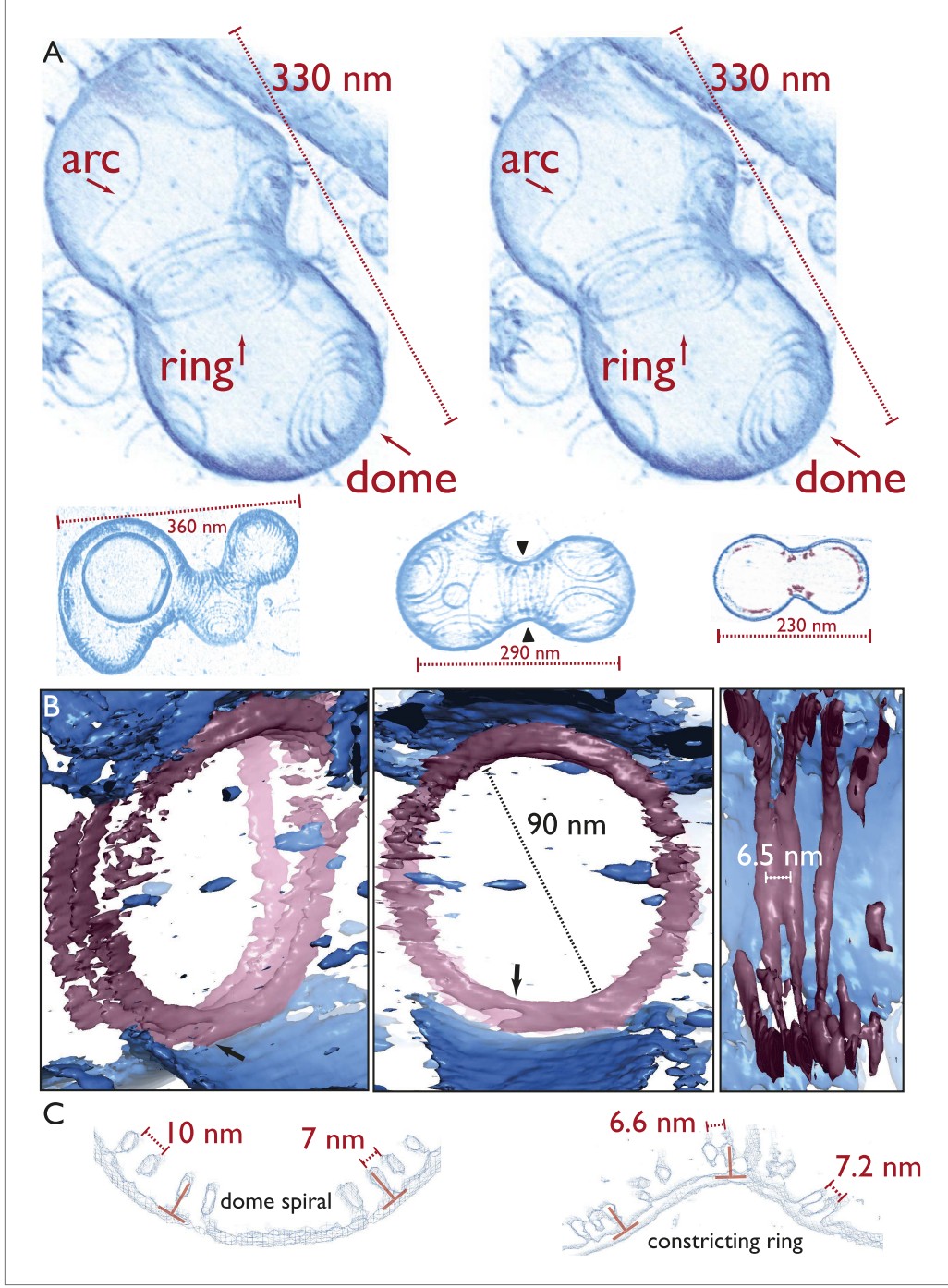

**Figure 4**. Electron cryotomography of liposomes constricted in vitro by rings of TmFtsA and TmFtsZ. (**A**) Stereo view of a representative liposome highlighting three different structures made by the enclosed TmFtsA and TmFtsZ proteins. Note that our images derived from tomographic volume data have not been segmented, they are volume representations of the actual 3D tomographic data. Arcs (also on the outside) are filaments made of both FtsA and FtsZ, whose curvature is determined by the mismatch in TmFtsA and TmFtsZ polymers subunit spacing (5 nm vs 4 nm, see also *Figure 3—figure supplement 1* & *Figure 4—figure supplement 2*). Dome-like structures are slightly helical spirals of condensing TmFtsZ filaments attached to the membrane by TmFtsA. Importantly, only complete rings seem capable of constriction force generation. The ring might consist of overlapping filaments (as in the stereo view and *Video 10*) or maybe a continuous helix of double filaments (bottom panel, middle liposome with black arrowheads, see also *Figure 4—figure supplement 1* and *Video 6*). The bottom panel
*Figure 4. Continued on next page*

*Figure 4. Continued*

depicts more examples of different liposome shapes and sizes. The cross-section (right) shows the distribution of filaments (red) inside a liposome (membrane in blue) (bottom right). *Video 4* shows a complete 3D volume in grey scale. *Video 5* shows a slice view at high magnification, demonstrating the excellent contrast these specimens generate, making it possible to see individual subunits and complete filament traces. *Videos 6–9* show 3D views of several constricted liposomes. *Figure 4—source data 1* enables 3D viewing of a liposome volume with PyMOL. (**B**) Close-up view of the FtsZ ring (purple) attached to the membrane (blue), here shown as single-threshold surface representations (these are not automatic or manual segmentations). The filaments overlap and interact laterally (left panel). View along the long axis shows that the ring is a perfect closed circle (middle panel). The black arrow points to where TmFtsZ and TmFtsA filaments are fully detached from each other. Individual filaments are resolved (right panel). *Video 10* shows a 3D walk-through the liposome, highlighting most features on the way. (**C**) Comparison of filament arrangements and geometries within the dome-like structures (left panel) and ring-like structures (right panel). Cross-sections demonstrate that in both cases, the TmFtsAZ filaments are positioned close to perpendicular with respect to the membrane (red symbols). However, the constriction force is generated only in the rings (see *Figure 5D* for explanation).

The following source data and figure supplements are available for figure 4:

**Source data 1**. PyMOL (version 1.7) session file showing volume and surface renderings of the liposome in stereo in Figure 4A, top.

**Figure supplement 1**. Constrictions occur only at the site of filament ring formation.

**Figure supplement 2**. A mechanism explaining variable intrinsic FtsA:FtsZ filament curvature.

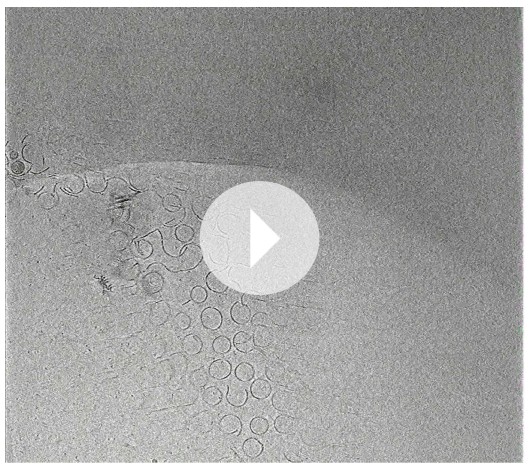

**Video 4**. This video shows a typical field of view from tomographic reconstruction of the in vitro reconstitution specimen. The filaments present on the water/air interface consist of TmFtsA and TmFtsZ filaments and therefore adopt a curved geometry. This corresponds to *Figure 4A*.

The rings were always closed, continuous without gaps, with some possibly consisting of one double filament forming a helix (*Figure 4—figure supplement 1* and *Video 6*) and others containing several shorter filaments in a helix-like arrangement (*Figure 4A* stereo view and *Video 10*). When the filaments were investigated end-on, they appeared to always be arranged close to 90° with respect to the liposome membrane tangent (*Figure 4C*), which was also observed for the filaments in cells (*Figure 2C,G*). All of these features are best demonstrated in *Video 10*, which provides an overview of the constriction and filament architectures and should be consulted to appreciate these findings properly.

## A semi-atomic model of the FtsZ ring constricting a liposome

Map quality allowed us to fit the FtsZ crystal structure manually and roughly through a spline curve and arrive at a pseudo-atomic model for an FtsZ ring (*Figure 5A*), making it possible to judge distances and dimensions relative to the crystal structures. A more detailed view using sphere representation (*Figure 5B*) shows, again, that the filaments within a ring were not exactly equidistant (black arrows) but came into direct contact only at certain points. Fitting the FtsA crystal structure into the map as well revealed two closely associated filaments and showed that the outline fit of the tomographic density is extremely good, although exact orientations and locations of the subunits along the filament of the molecules can only be guessed in most places given the resolution limit. It should be noted, though, that peaks appear in many places indicating the centre positions of individual FtsZ molecules (*Figure 5C*, *Figure 4–source data 1*, a PyMOL session file).

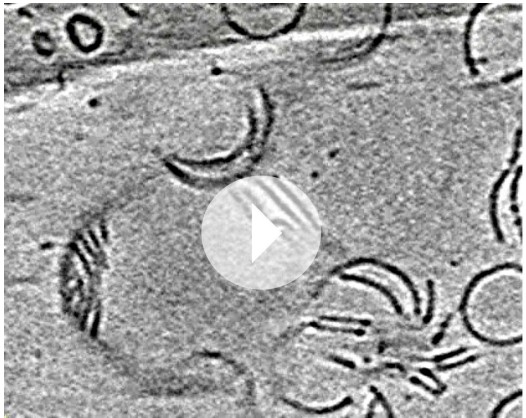

**Video 5**. This video shows a volume of the liposome whose stereo view is depicted in *Figure 4A*.

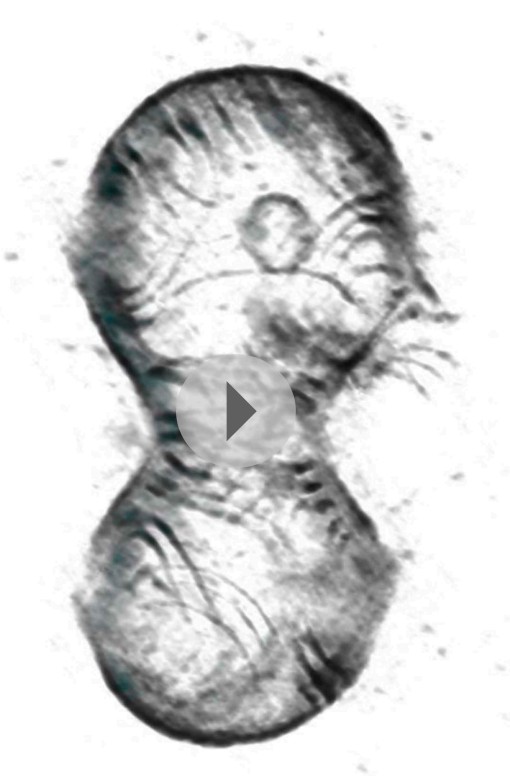

**Video 6**. This video shows a volume representation of the liposome that is depicted in *Figure 4A* (bottom middle panel, black arrowheads) and whose stereo view is shown in *Figure 4—figure supplement 1*.

## Discussion

### Constriction is accompanied by filament sliding

How do FtsZ and FtsA constrict liposomes? Is this likely to be related to the in vivo situation? Given that the filament architectures observed here in *C. crescentus* and *E. coli* and in constricting liposomes are so similar, we would suggest that the model we propose should be valid for both in vitro and in vivo, at least at some primordial level. FtsA forms (partial) filaments between the membrane and FtsZ filaments, and the filaments together encircle the constriction site while forming a single-layered small band of filaments. The entire structure is slightly helical, and shorter filaments overlap to form a continuous, closed ring.

By imaging completely unmodified cells, utilising recent advances in cryo-EM and acquiring tomograms of cells parallel to the tilt axis, we concluded that the FtsZ ring in cells is most likely continuous, probably made of shorter overlapping filaments. Previous analysis of *C. crescentus* cells by cryo-ET also showed that the FtsZ ring consists of overlapping filaments inside the inner membrane, although not all cells showed continuous rings (*Li et al., 2007*). Equally, results obtained with super resolution fluorescence microscopy techniques (*Holden et al., 2014*) showed punctuated fluorescence, possibly indicating non-continuous rings. We think it is important to point out that fluorescence microscopy only images the labelled species and intensity fluctuations within the ring may have arisen from using non-functional GFP fusions and/or their over-expression. Or fluctuations coming from overlapping filaments may have been over-emphasised during image analysis because of very low signal-to-noise. Unfortunately, currently, cellular tomography data are too weak to be able to trace individual filaments and their ends with confidence, so we have no direct evidence from our in vivo data for the length of individual FtsZ filaments making up continuous rings.

Given a continuous ring, at least in the in vitro situation, with no GTP turnover (or no added nucleotide) there is no filament shortening, meaning constriction requires the filaments to slide along each other as the ring decreases in diameter and the membrane deforms. That opposing forces from the filaments and the membrane surface are at work is most evident from the fact that the rings were always perfectly round, in contrast to the rest of the liposomes (*Figure 4B*, middle, *Video 10*).

In this context, the filament spirals that form shallow 'dome' structures in some of the liposomes and on flat monolayers (*Figures 3A and 4A*, top and *Figure 4—figure supplement 1*) are most revealing. They appear to be the result of sliding and condensation and show decreasing filament

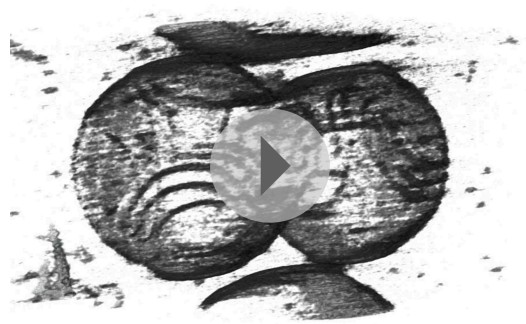

**Video 7**. This video shows the two remaining liposomes that are depicted in *Figure 4A*.

curvature, but do not deform the lipid membrane. This can be explained because the constriction force, which acts in the plane of the spirals, will not exert any force on the membrane since it is tangential (*Figure 5D*, right). In contrast, if the filaments form a ring around the volume of the liposome (in the middle, not at the poles), the constriction force is perpendicular to the membrane and will lead to the membrane being pulled in (*Figure 5D*, left).

Since in this model force generation is dependent on a closed ring, the system becomes self-regulating since constriction will only commence after a complete ring has formed. If the cell is too large or not enough FtsZ is available, constriction will not begin. It is important to note that only a closed, continuous ring is required, but it may consist of a number of shorter, overlapping filaments (as it did in the liposome reconstitutions, *Figure 4*).

## Where is the energy coming from for sliding and constriction?

What drives constriction of the closed rings and filament sliding? We propose three possible mechanisms that may even act in concert: (a) maximising filament overlap via sliding, (b) increasing repeat mismatch, and (c) repeated filament shortening through nucleotide turnover.

A. When the overlap between the filaments that are attracted to each other increases, more and more binding energy is produced. This has been proposed before to be theoretically sufficient for the constriction process (*Lan et al., 2009*) (*Figure 5E*). The lateral spacing of 6.5–8 nm between filaments we report here is slightly larger than the thickness of FtsZ filaments and presumably also FtsA filaments (*Matsui et al., 2012*; *Szwedziak et al., 2012*). Previous in vitro work reported an interfilament distance of 5 nm using FtsZ-mts; however, this was after negative staining and dehydration and no constrictions were observed, possibly due to lateral interactions being too tight (*Milam et al., 2012*). In *C. crescentus* cells, the lateral spacing between filaments was found to be 9.3 nm previously (*Li et al., 2007*)(*Li et al., 2007*) and we report here distances of ~7.8 (*Figure 1A*) in *C. crescentus* and ~6.8 nm (*Figure 1G*) in *E. coli*. AFM using only FtsZ, but observing spiral condensation, provided an even larger distance of 12 nm (*Mingorance et al., 2005*). All of these measurements are averages with large variances. One may conclude that the filaments in the FtsZ ring interact transiently and direct contact is localised to only a few small regions within the ring at a time. This could facilitate the constriction process since the filaments have to be free to slide. It was suggested previously that instead of forming many intermolecular solid bonds, which would lead to avidity and a barrier to sliding, an attractive force over a longer distance would keep the filaments apart while interacting (*Hörger et al., 2008*). This is more akin to the liquid state of matter, where many transient homotypic interactions, counteracted by thermal motion, lead to a fluid situation without absolute order but still keeping the molecules together.

B. The second possible driver of constriction comes from the repeat length mismatch of FtsA and FtsZ (*Szwedziak et al., 2012*). Although it is evident from our data that the curvature of individual filaments can not deform liposomes significantly from the inside, the decreasing diameter of the ring accompanied by increasing membrane curvature might enable more and more FtsA to be added, until an optimum curvature of the system has been achieved (*Figure 4—figure supplement 2*); this could provide additional energy and would also explain why FtsA exists at all and FtsZ is not directly attached to the membrane.

C. Why does FtsZ hydrolyse GTP then? We speculate that when constriction starts at large diameters (1 μm in *E. coli*), longer GTP-induced FtsZ filaments are needed to reach reliably around the cell in order to produce overlap for the ring and 'force engagement' to start the process. However, increasing overlap, developing as the constriction progresses, might lead to a kinetic barrier of sliding through avidity and the filaments would then have to be shortened. Formally this provides a third possible driver of constriction, at least for large constriction distances.

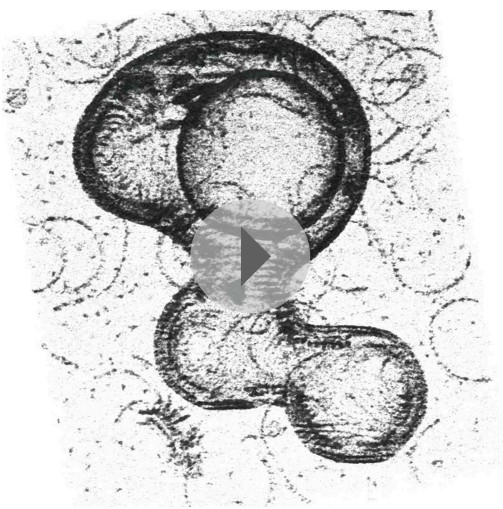

**Video 8**. This video shows the two remaining liposomes that are depicted in *Figure 4A*.

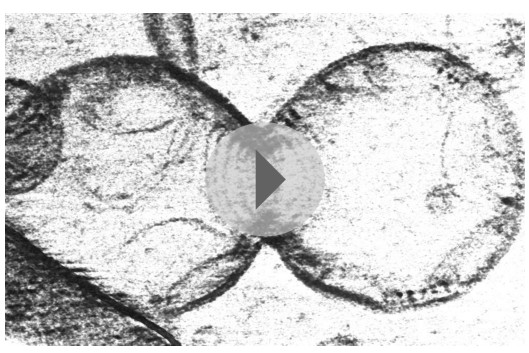

**Video 9**. This video shows a well-pronounced constriction with spirals being very prominent on lateral sides of the leading membrane edge, which eventually might lead to abscission. Not shown in any other figure. See also *Figure 5D*, middle for an explanation.

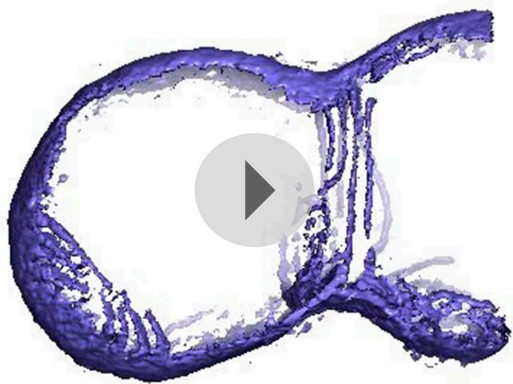

**Video 10**. This video runs through a surface representation of the liposome whose stereo view is depicted in *Figure 4A*, top, with features of interest highlighted along the way.

## Dynamics of the FtsZ ring are essential for its function

Continual depolymerisation and re-polymerisation through nucleotide turnover by FtsZ and/or FtsA, as shown in vivo by FRAP (*Stricker et al., 2002*), might also ensure that the ring never reaches a highly condensed state and FtsZ monomer-sequestering inhibitors such as SulA remain able to stop the process at any time (*Chen et al., 2012*). However, it has been reported that the GTP hydrolysis-deficient FtsZ(D212G) mutant generates prominent constrictions of tubular liposomes in vitro (*Osawa and Erickson, 2011*) and functions in cell division in *E. coli* (*Bi and Lutkenhaus, 1992*; *Trusca et al., 1998*; *Osawa and Erickson, 2006*). Furthermore, the use of non-hydrolysable GTP analogues did not impair the formation of condensed FtsZ structures on mica (*Hörger et al., 2008*). Taken together with our result that constriction of liposomes may in principle be independent of GTP hydrolysis, these data question the alternative idea of force generation by filament bending upon GTP hydrolysis as has been suggested previously (*Lu et al., 2000*; *Erickson et al., 2010*; *Li et al., 2013*). We suggest that nucleotide binding/hydrolysis is required solely for filament growth/shrinkage as these are essential to maintain the dynamic state of the FtsZ ring in cells.

Taken together, we envisage that in vitro liposome constrictions and in vivo cell division quite possibly utilise a different set of energetic drivers (a–c); for example, GTP hydrolysis was not required in vitro but clearly plays a role in vivo. And of course, it is likely that the cell wall synthesis in the periplasm, guided by the Z-ring through the divisome, provides additional force in cells. So far, wall-less bacterial L-forms have not shed light on this interdependence of cell wall synthesis and the FtsZ ring since artificial L-forms were found to divide by blebbing, most likely a mechanism that is not actively supported by cellular machinery (*Leaver et al., 2009*). FtsZ-based cell division is not functioning in these L-forms and we suggest this may be because L-forms are too large for Z-rings to close, given the amounts of FtsAZ present.

## How to perform abscission?

The described FtsZ filament arrangement might also provide a solution to the abscission problem: how do the membranes fuse at the end of division when the protein filaments are in between? *Figure 5D*, middle shows how an intermediate between the rings and the domes (as is present in the liposome constriction shown in *Figure 4A*,

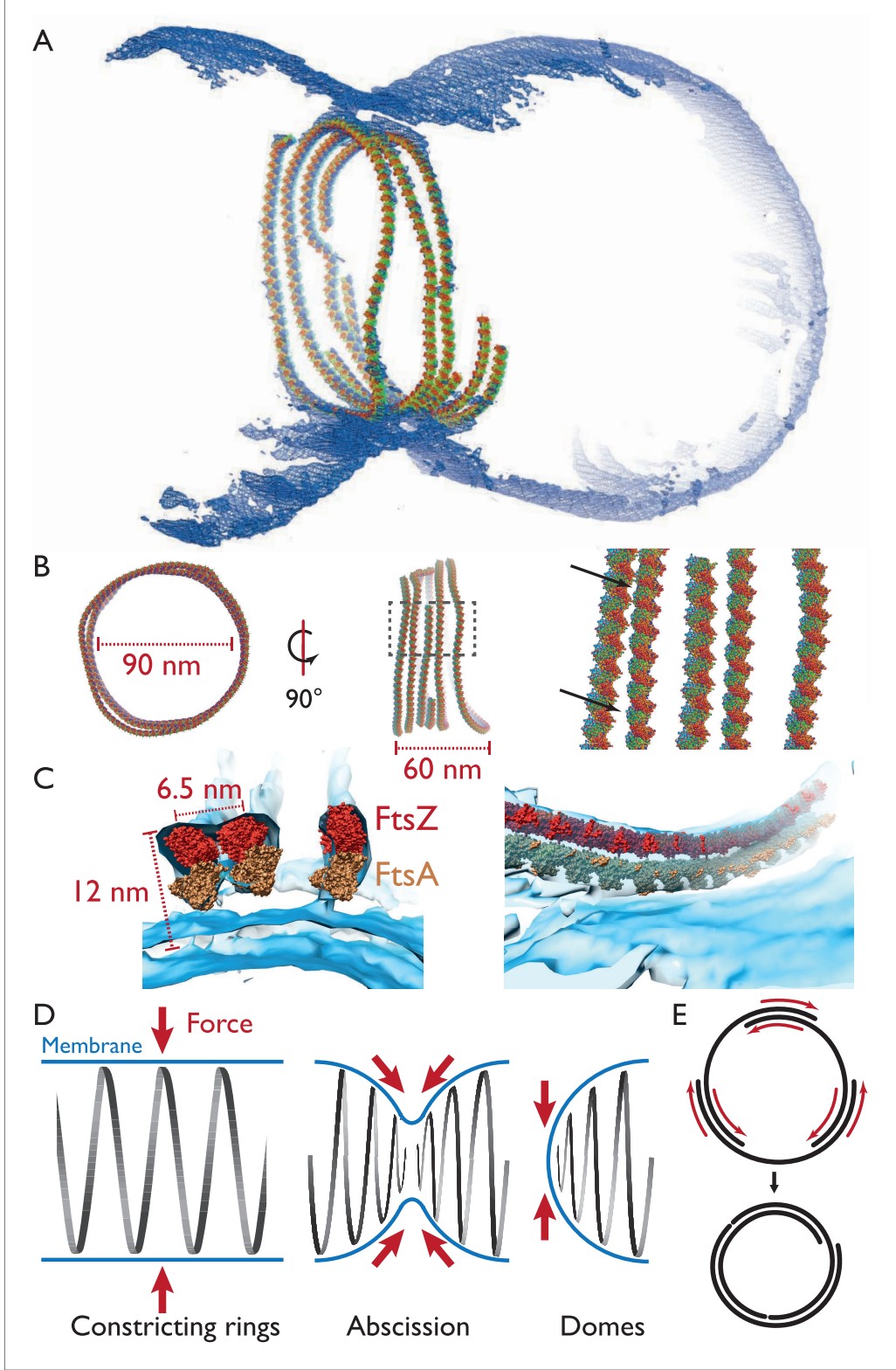

**Figure 5**. Visualising the FtsZ ring at the molecular level. (**A**) A semi-atomic model of the FtsZ ring constricting a liposome. 294 monomers of *S. aureus* FtsZ have been roughly positioned using a spline-fitting approach (PDB 3VO8 (*Matsui et al., 2012*)). This uses the same tomography data as *Figure 4A*. (**B**) The ring is 90 nm in diameter (left) and 60-nm thick (middle). It consists of at least four individual filaments (right, atoms shown as spheres) with

*Figure 5. Continued*

varying lateral interfilament distances (right, atoms shown as spheres, black arrows). (**C**) FtsZ filaments are single protofilaments, but they tend to pair in doublets. A precision manual fit of the TmFtsA polymer crystal structure (PDB 4A2B) (*Szwedziak et al., 2012*) in addition to 3VO8 FtsZ polymer crystal structure was performed in a region of very good density. The fit is excellent and dimensions and distances match well with CcFtsZ, EcFtsZ, and TmFtsAZ in vivo situations (*Figure 1A,E, 2E,G*). (**D**) Left: in the ring-like structures (black), force (red arrows) is perpendicular to the membrane (blue), leading to constriction. Middle: during constriction, the ring develops into two helical spirals, leading to forces pushing membrane inwards, and this might explain how abscission is accomplished since membranes will presumably not fuse while the protein filaments are in between (see *Figure 4A* bottom right and *Video 9* for an example of this in liposomes). Right: the domes we observed do not deform liposomes because the force generated is almost perfectly tangential to the membrane. (**E**) Constriction force generation and filament sliding. In the discussion, three different energy sources for constriction are listed: maximising filament overlap, repeat mismatch within FtsA–FtsZ copolymers (*Figure 4—figure supplement 2*) and filament shortening and turnover due to nucleotide hydrolysis by FtsAZ. While it is currently not obvious which of these or if a combination of the three mechanisms drives constriction, it seems clear to us that constriction, at least in the liposome reconstitution experiments, requires filaments to slide past each other as is depicted in two dimensions. Since also unmodified wild-type cells (*Figure 1*) show closed continuous rings at division sites, we would assume the same holds true in vivo. Filament sliding can also explain the spirals on lipid monolayers (*Figure 3A*) and spirals in the dome-like structures with liposomes (*Figure 4A*). The schematic drawn is a simplification into two dimensions, of course, in vivo and in vitro FtsZ filaments overlap in the third dimension, forming single-layered bands since each filament is anchored to the membrane.

bottom right and *Video 9*) may explain abscission, since the protein ring would normally be in the way of membrane fusion/fission at the end. The change from a flat band of filaments towards the helical spirals enables inward force to be developed on each side of the constriction, with a helical spiral on each side. The spirals observed in the dome-like structures might even be remnants of such liposome abscission events, although we have no evidence for this (*Figure 5D*, right). It remains to be seen if FtsZ is involved in final abscission since it has recently been reported that FtsZ might leave the septum earlier (*Söderström et al., 2014*).

## Similarities to other membrane remodelling systems

It is important to mention that membrane constriction with ESCRT-III and dynamin filaments has also been suggested to involve sliding helical filaments (*Roux et al., 2006*; *Guizetti et al., 2011*) and similar arrangements to those depicted in *Figure 5D*, right were predicted for the ESCRT-III system (*Fabrikant et al., 2009*).

Finally, our reconstitution of cell division can easily be adapted to include other cell division proteins, such as the division site selection mechanism MinCDE, nucleoid occlusion, FtsZ cross-linkers such as ZapA, and many more middle and outer divisome components.

## Materials and methods

### Plasmids and strains

Plasmids used in this work are listed in *Supplementary File 1A*. *E. coli* DH5α was used for cloning. *Caulobacter crescentus* NA1000/CB15N and *E. coli* B/r H266 (*Trueba and Woldringh, 1980*) were used for cellular electron cryotomography.

### Cellular tomography sample preparation

*Caulobacter crescentus* was grown overnight in PYE medium at 30°C. The overnight culture was used to inoculate 50 ml of M2G medium. The culture was grown at 30°C until the OD reached 0.5. 11 μl of this culture was mixed with 1 μl of protein-A conjugated to 10 nm gold beads (CMC, Leiden) and applied to freshly glow-discharged 300 mesh Cu/Rh Quantifoil (3.5/1) grids. The grids were plunge-frozen into liquid ethane using a FEI Vitrobot (Mark IV) and stored in liquid nitrogen.

*E. coli* cells (some containing relevant plasmids, for FtsZ mutant co-expression with endogenous wild-type FtsZ, *Supplementary file 1, Tables A and B*) were grown at 30°C in M9 minimal media supplemented with 0.4% glycerol until log-phase. Cells were then diluted into fresh M9 media with 0.02% arabinose (where needed, final concentration) and grown for 1–2 hr for FtsZ mutant protein expression.

## Lipid monolayer assay

*Thermotoga maritima* FtsZ (TmFtsZ) and FtsA (TmFtsA) proteins were purified and 2D monolayers were prepared as described previously (*Szwedziak et al., 2012*), taking extra care and verifying by ESMS that the C-terminal tails of both proteins were intact after purification as they are prone to proteolytic cleavage. This was not obvious from gels, sometimes.

## In vitro reconstitution of TmFtsZ and TmFtsA outside liposomes

20 µl of *E. coli* total lipid extract (Avanti Polar Lipids, Alabaster, AL) chloroform solution at 10 mg/ml was dried in a glass vial (Wheaton, Millville, NJ) under a stream of nitrogen gas and left overnight under vacuum to remove traces of the solvent. The resulting thin lipid film was hydrated with 200 µl of $TEN_{100}7.5$ buffer (50 mM Tris/HCl, 100 mM NaCl, 1 mM EDTA, 1 mM $NaN_3$, pH 7.5), containing either TmFtsA at 20 µM or TmFtsZ at 60 µM or both proteins. After 10 min of incubation at room temperature, the solutions were sonicated for 1 min in a water bath sonicator and then 2.5 µl of sample was plunge-frozen onto Quantifoil R2/2 holey carbon grids (Quantifoil, Germany) using an FEI Vitrobot (FEI Hillsboro, OR). Samples were stored in liquid nitrogen.

## In vitro reconstitution of TmFtsZ and TmFtsA inside liposomes

50 µl of *E. coli* total lipid extract chloroform solution at 10 mg/ml was dried in a glass vial under a stream of nitrogen gas and left overnight under vacuum to remove traces of the solvent. The resulting thin lipid film was hydrated with 50 µl of $TEN_{100}7.5$ plus 20 mM CHAPS (Anatrace, Maumee, Ohio) and shaken vigorously at 800 rpm using a benchtop Eppendorf shaker for 2 hr. The lipid–detergent solution was sonicated for 1 min in a water bath sonicator. Subsequently, 50 µl of TmFtsZ (30 µM) and TmFtsA (10 µM) solutions supplemented with 0.5 mM MgGTP or MgGMPCPP (Jena Bioscience, Germany) or no nucleotide was added and left for 30 min at room temperature. Next, the mixture was gradually diluted within 10 to 20 min to 600 µl with $TEN_{100}7.5$ or $TEN_{100}7.5$ plus nucleotides (both without detergent) to trigger spontaneous liposome formation. 2.5 µl of the solution was mixed with 0.2 µl 5 nm IgG immunogold conjugate (TAAB, UK) and plunge-frozen onto Quantifoil R2/2 holey carbon grid using an FEI Vitrobot.

## Electron cryomicroscopy and cryotomography

2D electron cryomicroscopy (cryo-TEM) images were taken on an FEI TECNAI Spirit TEM operating at 120 kV with a 2k × 2k CCD camera at a magnification of 42 k, corresponding to a pixel size of 0.25 nm. For electron cryotomography, samples were imaged using an FEI Polara or FEI Titan Krios TEM operating at 300 kV, equipped with a Gatan imaging filter set at zero-loss peak with a slit-width of 20 eV. A Gatan Ultrascan 4000 CCD camera binned to 2k × 2k or a 4k × 4k K2 Summit direct electron detector was used for data acquisition with SerialEM software (*Mastronarde, 2005*). Cells or in vitro reconstituted systems were imaged at a magnification of 41 k, corresponding to a pixel size of 5.8 Å (with US4000), or at a magnification of 26 k, corresponding to a pixel size of 4.5 Å (for K2) at the specimen level. Specimens were tilted from approximately −60° to +60° (±65° for *C. crescentus* cells) with a 1° increment. The defocus was set between 8 and 10 µm, and the total dose for each tilt series was around 120 e/Å$^2$ for in vitro reconstitution samples and 150–200 e/Å$^2$ for cells.

## Image processing

Tomographic reconstructions from tilt series were calculated using RAPTOR (*Amat et al., 2008*) and the IMOD tomography reconstruction package, followed by SIRT reconstruction with the PRIISM software or the TOMO3D package (*Chen et al., 1996*; *Kremer et al., 1996*; *Agulleiro and Fernandez, 2011*). Measurements of distances between structures were carried out within IMOD. Videos showing liposomes were prepared with PyMOL (*DeLano, 2002*).

## Structured illumination microscopy (SIM)

*E. coli* B/r H266 cells with plasmid pMZ124 were grown in LB medium at 30°C. At an $OD_{600}$ of 0.2, FtsZ(D212A) and FtsA expression was induced by adding 0.02% arabinose. After 2 hr, cell membranes were stained with FM4-64 membrane dye and cells were mounted on an agarose pad and visualised using a Nikon N-SIM microscope in the 2D-SIM mode.

## Western blot

FtsZ expression level in cells used for electron cryotomography experiments were examined with Western blots using rabbit anti-FtsZ primary antibodies (Agrisera, Sweden) and donkey anti-rabbit

IgG conjugated with horseradish peroxidase (GE Healthcare) and detected with ECL blotting reagent.

## Database deposition

The *Caulobacter crescentus* tomogram shown in *Figure 1A* has been deposited in the EM databank with accession number EMD-2814. The edited tomogram of TmFtsAZ constricting a liposome, as shown in *Figures 4A and 5A* and *Video 10*, has been deposited with accession number EMD-2815.

## Acknowledgements

We are very grateful to Leonid Sazanov (MRC Mitochondrial Biology Unit, Cambridge UK) for the suggestion to use CHAPS-solubilised lipids for liposome formation. We would like to thank Colin Palmer, Shaoxia Chen, and Christos Savva (MRC-LMB) for help with electron microscopy. Work on LMB's FEI Krios microscope was aided by Matthijn Vos and Sonja Welsch (FEI). Tom Goddard (UCSF) provided excellent help with spline fitting of the filaments. We would like to acknowledge access to a Nikon N-SIM microscope at the MRC Mitochondrial Biology Unit, Cambridge UK. This work was supported by the Medical Research Council (U105184326) and the Wellcome Trust (095514/Z/11/Z to JL).

## Additional information

### Funding

| Funder | Grant reference number | Author |
| --- | --- | --- |
| Medical Research Council | U105184326 | Piotr Szwedziak, Qing Wang, Tanmay A M Bharat, Matthew Tsim, Jan Löwe |
| Wellcome Trust | 095514/Z/11/Z | Piotr Szwedziak, Qing Wang, Jan Löwe |

The funders had no role in study design, data collection and interpretation, or the decision to submit the work for publication.

### Author contributions

PS, QW, Conception and design, Acquisition of data, Analysis and interpretation of data, Drafting or revising the article; TAMB, MT, Conception and design, Acquisition of data, Analysis and interpretation of data; JL, Conception and design, Analysis and interpretation of data, Drafting or revising the article

### Author ORCIDs

Piotr Szwedziak, http://orcid.org/0000-0002-5766-0873

## Additional files

### Supplementary file

• Supplementary file 1. (A) Plasmids used in this study. (B) Exact protein sequences of modified *E. coli* FtsZ proteins used for in vivo tomography experiments.

### Major datasets

The following datasets were generated:

| Author(s) | Year | Dataset title | Dataset ID and/or URL | Database, license, and accessibility information |
| --- | --- | --- | --- | --- |
| Szwediak P, Wang Q, Bharat TAM, Tsim M, Löwe J | 2014 | *Caulobacter crescentus* tomogram | http://www.ebi.ac.uk/pdbe/entry/EMD-2814 | Publicly available at the Electron Microscopy Data Bank. |

| Szwediak P, Wang Q, Bharat TAM, Tsim M, Löwe J | 2014 | Edited tomogram of TmFtsAZ constricting a liposome | http://www.ebi.ac.uk/pdbe/entry/EMD-2815 | Publicly available at the Electron Microscopy Data Bank. |

The following previously published datasets were used:

| Author(s) | Year | Dataset title | Dataset ID and/or URL | Database, license, and accessibility information |
|---|---|---|---|---|
| Matsui T, Yamane J, Mogi N, Yamaguchi H, Takemoto H, Yao M, Tanaka I | 2012 | *Staphylococcus aureus* FtsZ GDP-form | http://www.pdb.org/pdb/explore/explore.do?structureId=3vo8 | Publicly available at RCSB Protein Data Bank. |
| Szwedziak P, Wang Q, Freund SMV, Löwe J | 2012 | *Thermotoga maritima* FtsA with ATP gamma S | http://www.pdb.org/pdb/explore/explore.do?structureId=4a2b | Publicly available at RCSB Protein Data Bank. |

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
