## [Decision Letter]

Thank you for sending your work entitled “Architecture of the FtsZ ring in vivo
and in vitro” for consideration at *eLife*. Your article has been
favorably evaluated by Richard Losick (Senior editor), Werner Kühlbrandt (Reviewing
editor), and 3 reviewers, one of whom, Ethan Garner, has agreed to share his
identity.

The reviewers found that this paper gives some of the best images of FtsZ rings attained
thus far, both in vivo and in vitro. There are also number of findings of importance in
this work that could shift the current perception of how bacteria divide.

A number of the findings described in this paper have recently been published by others,
in particular the reconstitution of constriction of FtsA and FtsZ inside vesicles, and
spirals of FtsZ / FtsA on lipid bilayers. While the preliminary publications of these
results is unfortunate, and could be interpreted as stealing some of the authors’
“thunder”, the reviewers do not feel the authors should be penalized for
this, as their manuscript puts all these observations into a more coherent
interpretation than the previous studies, as well as showing high-resolution images.

The reviewers therefore find that with some modifications, in particular to the
discussion, this paper should be suitable for *eLife*. The Reviewing
editor has assembled the following comments to help you prepare a revised
submission.

1) The text states that “the filaments are ... uninterrupted everywhere the
missing wedge allows it”, but one reviewer noted filaments apparently terminating
inside the visible region (as assessed by the extent of the adjacent membranes) in Figure 1 (bottom), 1F (right bottom), 1H (bottom),
Figure 1—figure supplement 2 panels B
(right bottom) and D (left bottom), Figure 1—figure supplement 3 panels B (bottom) and E (left and right top),
Figure 1—figure supplement 4 panels B
(bottom). That is actually about half the filaments shown in the figures. This claim
should therefore be re-examined and clarified. The text further claims that the lack of
complete rings in several cells could be explained by their orientation with respect to
the tilt axis. A single sentence should be added simply listing the angular orientations
of the cells with and without apparently complete rings so readers can see how
consistent the trend is. Representative tomographic volumes should be deposited in a
suitable database upon publication.

2) Previous electron cryotomography from the Jensen lab and high-resolution fluorescence
imaging of GFP-FtsZ from at least 3 labs support an often-discontinuous Z ring. The
contrast between the authors result and the current literature view is hardly addressed
and should be discussed more fully.

3) The results for the in vitro reconstitution of FtsZ and FtsA in lipid vesicles are
quite beautiful and the quality of the images is impressive. The question is what they
mean; in other words does this reconstitution of FtsA and FtsZ filaments in vitro have
any significance to cell division in vivo? One of the reviewers doubts it, because there
may not be enough FtsA in the cell to make a continuous filament. Please address this
issue.

4) The authors' data supports an idea that “filament sliding” could
be powering division. This is a historic model in the field. The model presented by the
authors is slightly different from previous sliding models, and they should explain
these differences clearly.

5) The authors need to explain how their filament-sliding model agrees with the dynamics
that have been observed in vivo and in vitro. The authors state that the filaments
should slide against each other, and also acknowledge that the filaments turn over.

6) The authors propose that membrane constriction is driven by “repeated filament
shortening through nucleotide turnover”, and on first pass, this would appear to
fit with the rapid recoveries observed in vitro. But, on closer inspection, this becomes
hard to reconcile with other observations:

First, if there are very few long filaments in the cell, and filaments exchange at the
ends, how can the filaments constantly turn over at the rapid rate that has been
observed in vitro and in vivo?

Second, if the filaments are both long, and turn over at the ends, why do Z-rings
recover from all positions equally after photobleaching? If the filaments are long,
going around the cell, and recovering from polymerization/depolymerisation at the ends,
the rings should recover from the sides, not all over the bleached length.

7) How do the authors reconcile “sliding” with both the in vitro
observations of Loose (see “The bacterial cell division proteins FtsA and FtsZ
self-organize into dynamic cytoskeletal patterns”, Martin Loose and Timothy J.
Mitchison) and the single-molecule tracking of FtsZ (see “Investigating
Intracellular Dynamics of FtsZ Cytoskeleton with Photoactivation Single-Molecule
Tracking”, Lili Niu Ji Yu)? The in vitro loose work shows that the filaments
slide, but the monomers within filaments are immobile, i.e. the filaments are
treadmilling, not “sliding”. Similarly, photoactivated FtsZ has shown
monomers to be immobile in vivo, also in line with turnover by treadmilling, with static
monomers in the filament. The authors need to address these points.

8) Why do the authors assume the sliding must occur between protfilaments? Could it not
also occur between filament pairs?

9) The authors should discuss how simple protein crowding of non-filamentous proteins
alone can induce bending or tubulation (see “Steric confinement of proteins on
lipid membranes can drive curvature and tubulation”, Stachowiak JC1, Hayden CC,
Sasaki DY; “Membrane bending by protein-protein crowding.” Stachowiak JC1,
Schmid EM, Ryan CJ, Ann HS, Sasaki DY, Sherman MB, Geissler PL, Fletcher DA, Hayden CC).
This “bending” by protein crowding could give rise to constrictive forces
similar to (or in addition to) constriction resulting from FtsA-induced curvature. This
“crowding effect”, driven by depletion interactions would push the
filaments together. These depletion interactions could be very strong in vivo, due to
the crowded nature of the cell.

---

## [Author Response]

*1) The text states that “the filaments are ... uninterrupted everywhere
the missing wedge allows it”, but one reviewer noted filaments apparently
terminating inside the visible region (as assessed by the extent of the adjacent
membranes) in*
Figure 1
*(bottom), 1F (right bottom), 1H (bottom),*
Figure 1—figure supplement 2
*panels B (right bottom) and D (left bottom),*
Figure 1—figure supplement 3
*panels B (bottom) and E (left and right top),*
Figure 1—figure supplement 4
*panels B (bottom). That is actually about half the filaments shown in the
figures. This claim should therefore be re-examined and clarified. The text further
claims that the lack of complete rings in several cells could be explained by their
orientation with respect to the tilt axis. A single sentence should be added simply
listing the angular orientations of the cells with and without apparently complete
rings so readers can see how consistent the trend is. Representative tomographic
volumes should be deposited in a suitable database upon publication*.

For the *Caulobacter* cells, the quality of tomograms recorded and their
careful analysis in terms of cell orientation with respect to the tilt axis made it
possible to derive that the FtsZ ring is most likely continuous (it is the simplest
explanation for the data).

For *E. coli* the reviewers question if it is demonstrated, with the
figure panels listed, that it also contains a continuous ring. We agree that the
*E. coli* data is weaker (because the cells are much thicker). However
we strongly believe that *E. coli* also contains a continuous ring, as
the architecture of the Z-ring between two species of gram-negative bacteria sharing the
same cell division proteins would be expected to be retained. We would argue that the
rings sometimes appear to be non-continuous in WT *E. coli* tomograms
simply because the tomograms and images are not as good as for
*Caulobacter* (much more noise, as is obvious from the images) and
this was in fact the reason why we employed *Caulobacter* in the first
place.

Please also note in this context that when over-expressed, the Z-ring is definitely
continuous and these septa lead to normal divisions as far as we can tell.

To soften our conclusions somewhat, we have added the following text: “Based on
these observations we concluded that *E. coli* Z-rings were, like in
*C. crescentus*, probably continuous and consisted of single-layered
bands that are 5-10 filaments wide.”

Figure 1—figure supplement 1 Panel C
shows the effect of different orientations of cells with respect to the tilt axis and
the reader is referred to this section in the main text.

As for database submission, we have deposited the following two most important datasets
from our work, EMDB numbers now also included in the manuscript: EMD-2814 –
Caulobacter tomogram, Figure 1; EMD-2815
– FtsAZ liposome reconstitution tomogram, Figures 4 and 5, Video 10.

*2) Previous electron cryotomography from the Jensen lab and high-resolution
fluorescence imaging of GFP-FtsZ from at least 3 labs support an often-discontinuous
Z ring. The contrast between the authors result and the current literature view is
hardly addressed and should be discussed more fully*.

A paragraph addressing these issues has been added: “By imaging completely
unmodified cells, utilising recent advances in cryo-EM and acquiring tomograms of cells
parallel to the tilt axis we conclude that the FtsZ ring is most likely continuous, made
of shorter overlapping filaments. Previous analysis of *C. crescentus*
cells by cryo-ET also showed that the FtsZ ring consists of overlapping filaments inside
the inner membrane, although not all cells showed continuous rings {[27], EMBO J, 26, 4694-708}. Equally,
results obtained with super resolution fluorescence microscopy techniques (19) showed punctuated
fluorescence, possibly indicating non-continuous rings. We think it is important to
point out that fluorescence microscopy only images the labelled species and intensity
fluctuations within the ring may have arisen from using non-functional GFP fusions
and/or their over-expression. Or fluctuations coming from overlapping filaments may have
been over-emphasised during image analysis because of very low
signal-to-noise.”

*3) The results for the in vitro reconstitution of FtsZ and FtsA in lipid
vesicles are quite beautiful and the quality of the images is impressive. The
question is what they mean; in other words does this reconstitution of FtsA and FtsZ
filaments in vitro have any significance to cell division in vivo? One of the
reviewers doubts it, because there may not be enough FtsA in the cell to make a
continuous filament. Please address this issue*.

We do think that the *in vitro* architecture is similar and relevant to
what happens *in vivo*. The fact that there is not enough FtsA to make a
continuous filament (at least at the onset of the constriction) is part of our model
where a continuous FtsAZ co-filament is possible only at later stages where the membrane
curvature allows it (Figure 4—figure supplement 2). FtsZ and FtsA will co-polymerise, until the repeat mismatch makes that
energetically unfavourable at a given bending angle (curvature). This 'desire'
to continue to polymerise leads to further bending and we would propose that there will
first be short stretches of FtsA that then elongate when fully constricted (panel 1F
might show a bit of FtsA filaments, actually, since there are additional filaments at 8
nm distance from the membrane).

*4) The authors' data supports an idea that “filament sliding”
could be powering division. This is a historic model in the field. The model
presented by the authors is slightly different from previous sliding models, and they
should explain these differences clearly*.

We are not sure which paper the reviewers refer to. The sliding model is at length
discussed in (13) and this is/was
cited so we do not have to discuss all the various historical models at length.

*5) The authors need to explain how their filament-sliding model agrees with the
dynamics that have been observed in vivo and in vitro. The authors state that the
filaments should slide against each other, and also acknowledge that the filaments
turn over*.

Yes, we propose that the FtsZ ring made of short overlapping filaments stays in
accordance with the dynamics that have been observed by others, as this will enable the
filaments to grow and shrink. Already discussed in the manuscript text. Filaments are
free to depolymerise/polymerise as long as the ring remains continuous (because
otherwise no force can be applied to the membrane). This is a self-selecting principle,
of course, and one of the attractive features of our model. A whole paragraph in the
Discussion section is devoted to this problem and has been slightly modified to increase
clarity.

*6) The authors propose that membrane constriction is driven by “repeated
filament shortening through nucleotide turnover”, and on first pass, this
would appear to fit with the rapid recoveries observed in vitro. But, on closer
inspection*, *this becomes hard to reconcile with other
observations:*

*First, if there are very few long filaments in the cell*, *and
filaments exchange at the ends, how can the filaments constantly turn over at the
rapid rate that has been observed in vitro and in vivo?*

*Second, if the filaments are both long, and turn over at the ends, why do
Z-rings recover from all positions equally after photobleaching? If the filaments are
long, going around the cell, and recovering from polymerization/depolymerisation at
the ends, the rings should recover from the sides, not all over the bleached
length*.

We do not envisage that there are long filaments going round cells. Unfortunately, the
cellular tomograms do not have enough resolution to elucidate this with confidence; when
looking along the long cell axes, several filaments are viewed in projection so starts
and ends cannot be determined. And in the perpendicular orientation, signal is
definitely too low to identify individual filaments.

We propose that there might be many short overlapping filaments forming a ring,
producing many ends, enough for the dynamics to appear continuous. Recovery of ring-sub
fragments after bleaching may also happen because of sliding and many ends of short
filaments enable recovery everywhere.

*7) How do the authors reconcile “sliding” with both the in vitro
observations of Loose (see “The bacterial cell division proteins FtsA and FtsZ
self-organize into dynamic cytoskeletal patterns”, Martin Loose and Timothy J.
Mitchison) and the single-molecule tracking of FtsZ (see “Investigating
Intracellular Dynamics of FtsZ Cytoskeleton with Photoactivation Single-Molecule
Tracking”, Lili Niu Ji Yu)? The in vitro loose work shows that the filaments
slide, but the monomers within filaments are immobile, i.e. the filaments are
treadmilling, not “sliding”. Similarly, photoactivated FtsZ has shown
monomers to be immobile in vivo, also in line with turnover by treadmilling, with
static monomers in the filament. The authors need to address these
points*.

We think it is correct to state that filament treadmilling cannot lead to constriction
(because subunits do not move), unless end-tracking is involved, which no known
component in the system performs. So we think it is telling that in the mentioned work
no constriction was observed, in contrast to the work presented here.

The *in vivo* tracking is a difficult experiment and, again,
fluorescently labelled FtsZ was used. It may well be that that protein was
non-functional (and this is what was tracked) but we agree that similar experiments need
to be investigated in the future. We plan to do this with our liposomes, where small
chemical dyes may be employed, mitigating problems with large fusion proteins.

*8) Why do the authors assume the sliding must occur between protfilaments? Could
it not also occur between*
*filament pairs?*

Yes, this is possible and *in vivo* we often observed filaments that were
associated into doublets and the sliding could of course occur between such filament
pairs. This has not been discussed further because it does not change the model and
requires more knowledge about FtsZ/FtsA filament doublet formation.

*9) The authors should discuss how simple protein crowding of non-filamentous
proteins alone can induce bending or tubulation (see “Steric confinement of
proteins on lipid membranes can drive curvature and tubulation”, Stachowiak
JC1, Hayden CC, Sasaki DY; “Membrane bending by protein-protein
crowding.” Stachowiak JC1, Schmid EM, Ryan CJ, Ann HS, Sasaki DY, Sherman MB,
Geissler PL, Fletcher DA, Hayden CC). This “bending” by protein
crowding could give rise to constrictive forces similar to (or in addition to)
constriction resulting from FtsA-induced curvature. This “crowding
effect”, driven by depletion interactions would push the filaments together.
These depletion interactions could be very strong in vivo, due to the crowded nature
of the cell*.

This is a very attractive idea. However, it is not clear if this is still valid for
proteins acting on negatively curved membranes (e.g., inside cells/liposomes). Also, if
this was the case then FtsA (which is the membrane anchor here) itself would trigger
significant membrane deformations from the outside and, at least in our hands, one
really needs FtsZ to observe the tubulation. No action taken since there are many
theoretical studies published with many different ideas, which makes this more suitable
for a future review, we would suggest.